# Prediction of the Neurotoxic Potential of Chemicals Based on Modelling of Molecular Initiating Events Upstream of the Adverse Outcome Pathways of (Developmental) Neurotoxicity

**DOI:** 10.3390/ijms23063053

**Published:** 2022-03-11

**Authors:** Domenico Gadaleta, Nicoleta Spînu, Alessandra Roncaglioni, Mark T. D. Cronin, Emilio Benfenati

**Affiliations:** 1Laboratory of Environmental Chemistry and Toxicology, Department of Environmental Health Sciences, Istituto di Ricerche Farmacologiche Mario Negri IRCCS, Via Mario Negri 2, 20156 Milan, Italy; alessandra.roncaglioni@marionegri.it (A.R.); emilio.benfenati@marionegri.it (E.B.); 2School of Pharmacy and Biomolecular Sciences, Liverpool John Moores University, Byrom Street, Liverpool L3 3AF, UK; n.spinu@ljmu.ac.uk (N.S.); m.t.cronin@ljmu.ac.uk (M.T.D.C.)

**Keywords:** molecular initiating events, neurotoxicity, adverse outcome pathways, QSAR

## Abstract

Developmental and adult/ageing neurotoxicity is an area needing alternative methods for chemical risk assessment. The formulation of a strategy to screen large numbers of chemicals is highly relevant due to potential exposure to compounds that may have long-term adverse health consequences on the nervous system, leading to neurodegeneration. Adverse Outcome Pathways (AOPs) provide information on relevant molecular initiating events (MIEs) and key events (KEs) that could inform the development of computational alternatives for these complex effects. We propose a screening method integrating multiple Quantitative Structure–Activity Relationship (QSAR) models. The MIEs of existing AOP networks of developmental and adult/ageing neurotoxicity were modelled to predict neurotoxicity. Random Forests were used to model each MIE. Predictions returned by single models were integrated and evaluated for their capability to predict neurotoxicity. Specifically, MIE predictions were used within various types of classifiers and compared with other reference standards (chemical descriptors and structural fingerprints) to benchmark their predictive capability. Overall, classifiers based on MIE predictions returned predictive performances comparable to those based on chemical descriptors and structural fingerprints. The integrated computational approach described here will be beneficial for large-scale screening and prioritisation of chemicals as a function of their potential to cause long-term neurotoxic effects.

## 1. Introduction

The human brain is exceptionally sensitive to injury, and several neurodevelopmental processes have been shown to be highly vulnerable to external factors [1,2]. Such processes include neural progenitor cell proliferation, apoptosis, cell migration, neuronal and glial differentiation, neurite outgrowth and branching, myelination, synaptogenesis and neuronal network formation, the ontogeny of neurotransmitters and receptors, the development of the blood–brain barrier, and the developmental changes in the adolescent brain [3,4,5]. Disruption of any of these processes may lead to potentially adverse alterations in neuroanatomy, neurophysiology, and neurochemistry. It has been estimated that developmental neurotoxicity (DNT) disorders affect 10–15% of all births [6], and the prevalence of autism and attention-deficit hyperactivity disorders is increasing worldwide [1]. In addition, DNT disorders documented in children and adolescents could be a precursor of the development of neurodegenerative diseases (NDs) later in life [7]. NDs (e.g., Alzheimer’s and Parkinson’s) are widely investigated pathologies due to the low efficacy of current therapies [8,9], the severe functional impairments they impose on daily life activities, and the resulting high familial, social, and financial costs of patient care [10].

Overall, genetic factors seem to account for about 30–40% of all cases of DNT disorders [11]. Evidence has been reported that exposure to chemical stressors, e.g., industrial chemicals in the environment, is a key determinant in the occurrence of neurological disorders [11,12]. Thousands of chemicals have been reported to have adverse effect on neurodevelopment or to be toxic the nervous system in adults [11,12]. However, the total number of known neurotoxic substances is likely to be an underestimation of the true number released into the environment [11]. There is, therefore, a need to develop strategies able to screen the large number of chemicals to which the population is exposed daily and that may have possible long-term adverse health consequences to the brain. Guideline-based DNT studies involve the use of a large number of animals for an extended period of time, making this kind of study significantly resource-intensive and not suitable for large-scale screening [13,14].

Computational toxicology has been shown to be a cost- and time-efficient alternative to traditional toxicity testing methods [15]. Quantitative Structure–Activity Relationship (QSAR) models are computational methods that have been primarily applied for their capability to identify the toxicity of chemicals as a function of their structural attributes [16]. The increased availability of data obtained from in vitro bioactivity testing has made the development of QSARs easier. Moreover, QSARs require relatively few resources and are rapid, which have been key factors in their increased use assisting in filling toxicological data gaps for chemicals with high production volumes.

Toxicology has recently undergone a paradigm shift towards the use of alternative testing methods based on knowledge of the biological modes of action and pathways that are responsible for adverse effects, defined as adverse outcome pathways (AOPs). An AOP is a logical construct that connects an upstream molecular initiating event (MIE) (e.g., the interaction of a chemical with a molecular target) to a downstream adverse outcome (AO), progressing through a series of key events (KEs) [17,18]. According to this concept, compounds of unknown hazards can be assigned to various levels of concern based on the number of activated MIEs and the extent of their activation. Moreover, chemicals activating similar MIEs/KEs with respect to known toxic chemicals will be more likely to be toxic themselves [19]. Several authors have recently highlighted a possible synergism between the AOP concept and QSAR modelling in toxicology [19,20,21,22,23,24,25,26]. Thus, it is possible to utilise QSAR models to predict the potential of chemicals to modulate MIEs and to prioritise chemicals as a function of their toxicological profile.

In the present manuscript, we present an integrated computational system to predict neurotoxic potential that relies on the identification of the MIEs activated by chemicals. MIEs upstream of neurotoxicity were identified from recently published AOP networks, then QSAR models were developed for the prediction of each MIE. Predictions from QSARs for individual MIEs were evaluated for their capability to discriminate neurotoxic and non-neurotoxic compounds as part of an integrated computational prediction system. MIEs were used as independent variables in various machine learning approaches and compared for their predictive power with other widely used methods for chemical description, i.e., fingerprints and molecular descriptors. The predictions returned by QSARs presented here may represent an effective first tier of an Integrated Approaches to Testing and Assessment (IATA) to rapidly screen many chemicals, providing information regarding potential MIEs and associated mechanisms of toxicity and thus, helping to prioritise chemicals for additional and better-targeted screening/testing, e.g., in vitro testing [27,28].

## 2. Results

Performance statistics from external validation for the QSARs for MIEs are reported in Table 1. A complete description of the MIEs listed in Table 1 and of their abbreviations is reported in Section 4.1.

The statistics of the balanced random forest (BRF) models for MIEs were extremely good and confirmed the high quality of the information included in ChEMBL database and its suitability as a source of data for modelling the interactions of ligands with molecular targets upstream of biological pathways (e.g., nuclear receptors and enzymes) [29,30]. One possible reason for this high performance is the high structural homogeneity of the active samples. This is not surprising, as many records included in ChEMBL are congeneric sets of candidate drugs, while negative samples in the MIE datasets are more structurally heterogeneous. This aspect is reasonable, as ligands for enzymes and receptors are often required to possess specific pharmacophoric features to interact with binding sites, leading to preferred structural moieties being shared among the different ligands.

External validation returned BA values in the range of 0.84–0.99. The statistics were balanced between sensitivity (SEN) and specificity (SPE); although SEN was higher in most cases, slightly lower values for the inactive class were most common in all datasets. As for Matthews Correlation Coefficient (MCC) values, several of the models showed values below the average, such as the BRFs for CYP2E1, KAR, and TTR. MCC has beenproposed for the evaluation of classification between two very unbalanced categories; however, it was observed that this parameter can sometimes be biased by high unbalance datasets on terms of the categories (i.e., fewer than 20% of chemicals included in the smallest category) [24]. Indeed, the datasets mentioned above are among those with a lower number of actives.

Table 2 shows statistics for the neurotoxicity-predicting models. The performance of each classifier was calculated as the average of 100 iterations of five-fold cross validation. Figure 1 (Figure 1a: k-nearest neighbours (k-NN); Figure 1b: random forest (RF); Figure 1c: neural network (NNET)) shows the distribution of the BAs of models based on MIE predictions compared with DRAGON descriptors and extended fingerprints. In the case of k-NN classifiers, the MIE predictions show higher performance (BA avg = 0.72) with respect to chemical descriptors (BA avg = 0.70), and fingerprints do not perform as well. MIEs and descriptors have a similar peak in the distribution of BAs between 0.70–0.75. RFs are the top-performing classifiers overall with respect to kNN and NNET, always reaching average BAs higher than 0.65 and having maximum values closer to 0.90. In this case, DRAGON descriptors were the top-performing variables (BA avg = 0.83), followed by predictions of QSARs based on fingerprints (BA avg = 0.74) and MIEs (BA avg = 0.73). In the case of NNET, MIE predictions and DRAGON descriptors (BA avg = 0.74) were characterised by an almost equal distribution profile, while fingerprints had lower performance (BA avg = 0.67).

The relative importance of MIEs for neurotoxicity prediction was evaluated using the methods described in Section 4.6. Appendix A reports the impact of the removal of specific MIEs on the performance (BAs) of models for neurotoxicity, while Appendix A sorts the various MIEs included in the RF models for neurotoxicity by their variable importance. Both BAs and variable importance are an average of the values calculated over the various modelling iterations; see Section 4.5.

Thyroid elements seem to be relevant (THRs, TPO and TTR) to neurotoxicity; in particular, the exclusion of TTR always leads to a reduction in BA average. TTR is the fourth MIE in terms of variable importance. THRs, TPO and TTR are involved in the biosynthesis, metabolism, and transportation of the thyroid hormone, respectively. Among the two isoforms of thyroid receptors, THRβ is consistently the most important (first descriptor for variable importance), while the removal of THRα did not negatively affect performance. Among ionotropic glutamate receptors, AMPAR and KAR seem to be more linked to neurotoxicity than NMDAR. AMPA/kainite receptor-mediated neurotoxicity was reported to possibly play a role in neuronal neurodegeneration in amyotrophic lateral sclerosis [31] and in the injury of basal forebrain cholinergic neurons in diseases such as Alzheimer’s [32]. VGSC and, to a lesser extent, GABAR are consistently relevant within models developed for neurotoxicity (Appendix A). The relevance of VGSC and GABAR is confirmed in the RF variable importance analyses (Appendix A). The role of SMARTS for protein adduct formation is unclear: the descriptor is the least relevant from the variable importance analyses, and on the whole, its exclusion does not affect the average performance of RFs. On the other hand, its removal has a detrimental effect on the performance of both the K-NN and NNET models.

## 3. Discussion

In the present work, a new integrated computational system was proposed for predicting the neurotoxic potential of chemicals as a function of their capability to trigger MIE (i.e., interaction and modulation of relevant receptors and enzymes) upstream of neurotoxicity. QSARs were developed to predict MIE induction while BRFs were applied to handle the unbalanced training data. In the last part of the manuscript, MIE predictions were used to classify neurotoxic and non-neurotoxic compounds and compared for their predictivity with other approaches to describe the structure of chemicals, namely, chemical descriptors and extended fingerprints.

Overall, in two out of three cases MIEs perform comparably with chemical descriptors and better than fingerprints, which are considered the gold standard for describing chemical structures within QSARs [33]. The only exception is given by neurotoxicity models based on RF, where DRAGON descriptors (BA avg = 0.83) performed better than their counterparts based on MIEs (BA avg = 0.73). This is likely to be due to the fact that RFs perform better if trained on a larger pool of variables, such as the pool of descriptors provided by DRAGON (i.e., several thousand) [34,35].

Despite this, one of the key advantages afforded by the use of MIE responses in place of the classical structural representation of molecules is the interpretability of the predictions. Indeed, a complete profile of the neuronal receptors and enzymes that are activated is given together with the overall neurotoxicity outcome, providing insights into the possible mode of action of a predicted neurotoxic chemical. This aspect was further verified by predicting a series of known neurotoxicants with the models predicting MIEs. The mode of action for these neurotoxicants was reported in a review by Masjosthusmann and coworkers [36] who gathered information from the literature about the targets upstream of the neurotoxicological pathway of these chemicals [11,12,37]. Interestingly, QSARs for the prediction of MIEs were able to identify the correct mode of action of several neurotoxicants. For example, dichlofenthion (97-17-6), Parathion (56-38-2), paraoxon (311-45-5), diazinon (333-41-5), physostigmin (57-47-6), ibogaine (83-74-9), and dichlovoros (62-73-7) were correctly predicted to stimulate cholinergic neurotransmission through AChE inhibition, while 3-Nitropropionic acid (504-88-1), glyphosate (1071-83-6), and argiopine (105029-41-2) were predicted to interact with NAMDR receptors. Indeed, the two former chemicals are reported to stimulate glutamatergic neurotransmission and cause excitotoxicity after activation of NMDA, leading to oxidative stress and cell death, while the latter was reported to inhibit glutamatergic neurotransmission after blockage of the post-synaptic receptors. Rotenone (83-79-4) and dieldrin (60-57-1) were correctly predicted to inhibit complex I (NADH dehydrogenase) and to cause reactive oxygen species (ROS)-induced degeneration of dopaminergic neurons and locomotor deficit.

In addition to increased biological relevance, MIE predictions simplify models to a reduced number of variables (i.e., fewer than 20), while in the case of descriptors and fingerprints several hundred variables may be included in the models.

The RFs developed here for the prediction of single MIEs returned satisfactory predictive performance and were confirmed to be a valuable method in the field of computational toxicology. The statistical performance of the models presented here confirmed our previous findings that using RFs with internal balancing of categories (BRF) represents one of the best methods for handling the unbalanced distribution typical of biological data [24,38].

In the present manuscript, the use of biological information (i.e., MIEs) instead of the classical structural description of molecules was proposed for the development of QSARs. The use of biological data (e.g., biological assays) utilised as input variables to develop QSARs has been increasingly explored in the recent literature [39]. This strategy is justified by the fact that QSARs historically had difficulty predicting complex systemic endpoints encompassing several mechanisms, which are difficult to model together. In the case of neurotoxicity, the brain is an extremely complex organ comprising a variety of highly specialised neuronal cell types that differ in function, expression of brain regions, and stages of development [40]. These different cells are all potential targets that can be disrupted by neurotoxicants with different possible mechanisms of toxicity [4]. Another limitation of QSARs is that they rely on the principle that analogies in chemical structure always result in analogies in toxicity. However, the existence of activity cliffs, i.e., compounds with high structural similarity together with unexpectedly high activity differences, were reported for high-tier endpoints characterised by multiple mechanisms of toxicity [41]. On the other hand, the use of information from AOPs and biological assays allows for the fragmentation of complex endpoints into simpler ones based on mechanistic knowledge. These “sub-endpoints” are easier to address with a single computational model, as they describe the interaction of a chemical with a single molecular target that triggers a specific response. Overall, this strategy allows for a reduction in the complexity of the challenge of capturing the complex relationships existing between the structure of a chemical and its high-level systemic toxicity [4].

The development of new machine learning and artificial intelligence-based approaches is highly desirable, as it allows for the detection of chemicals with potential neurotoxicity and DNT effects in a more time- and resource-efficient way compared to traditional in vivo testing. In addition, data from in silico screenings based on AOPs can provide a scientifically sound rationale to make decisions relating to assessment of the safety of chemicals. The mechanistic nature of AOPs provides knowledge to guide the design of new IATAs to meet regulatory needs [28,42]. In particular, these in silico predictions can be used to provide information regarding the potential MIEs of chemicals, to help prioritise or deprioritise certain chemicals for further testing, and to provide indications for better-targeted follow-up in vitro evaluations [43]. In the specific case of neurotoxicity, a wide range of in vitro tests has been proposed, each evaluating a different MIE/KE of the complex network upstream of the adverse outcome [27]. Predictions of MIE provided by QSARs may give indications of which assays to prioritise among the wide battery of tests available. In this regard, in silico models represent an ideal first tier of a multi-step IATA for the prediction of the neurotoxicity of chemicals which involves multiple alternative testing methods.

## 4. Materials and Methods

### 4.1. Data Selection for Molecular Initiating Events (MIEs)

MIEs linked to neurotoxicity were identified from the AOP networks [44] published by Spînu et al. [45] and Li et al. [27]. The MIEs selected along with their associated molecular targets (i.e., receptors, enzymes) are reported in Table 3. In certain cases, multiple molecular targets are associated with a single MIE (e.g., three different glutamate receptors were considered for MIE A), while a single target may be repeated in multiple MIEs (e.g., NADHOX was common to MIEs C and N). The molecular targets involved in the MIEs and their relevance in neurotoxicity are briefly described below.
Glutamate ionotropic receptors, i.e., N-methyl-D-aspartate (NMDAR), alpha-amino-3-hydroxy-5-methyl-4-isoxazolepropionate (AMPAR) and kainate (KAR) are responsible for excitatory synaptic transmission and synaptic plasticity, which are fundamental for learning and memory [46]. Sustained over-activation of these receptors (MIE A) can induce excitotoxicity due to increased Ca^2+^ influx, with consequent cell death, memory problems, and convulsions [4]. Analogously, the chronic blockage of NMDAR by chemicals during synaptogenesis (MIE B) disrupts neuronal network formation, resulting in the impairment of learning and memory processes [47] and increasing the risk of developing Alzheimer’s-type NDs in later life [2].Protein adduct formation is the covalent interaction between an electrophilic chemical and the nucleophilic part of a protein, and may lead to damage of the protein and the potential loss of its function. This may affect thiol- and seleno-containing proteins, which offer antioxidant protection [48]. The binding of xenobiotics (e.g., heavy metals and mercury) to these or other proteins during brain development (MIE D and H) may lead to several functional impairments, such as in learning and memory. Cytochrome P450 2E1 (CYP2E1) is relevant to this mechanism as well, as it is one of the enzymes responsible for the metabolism of small compounds. The induction of CYP2E1 (MIE E) leads to an increase in reactive metabolites, which can form protein adducts. For example, a high concentration of ethanol leads to an increased expression of CYP2E1 and consequent increased production of acetaldehyde metabolite, which can form protein adducts [49]. The consequences include oxidative stress, lipid peroxidation, unfolded protein responses and, ultimately the apoptosis of neuronal cells [50].The function of the Na^+^/I^−^ symporter (NIS) is critical for the physiological production and maintenance of thyroid hormone levels in the serum, as it mediates the transport of iodide into thyroid cells. Its inhibition (MIE F) results in decreased thyroid hormone synthesis, with effects on neurocognitive function in children [51,52].Acetylcholinesterase (AChE) is an enzyme present in both central and peripheral nervous systems and in muscular motor plaques. It is responsible for the enzymatic cleavage of the neurotransmitter acetylcholine [53]. Inhibition of AChE (MIE I), e.g., by organophosphates and carbamates, leads to an increase in levels of acetylcholine and overstimulation of both muscarinic and nicotinic receptors, resulting in multiple adverse outcomes affecting a wide variety of functions [54].Ryanodine-sensitive Ca^2+^ channels (RyR) contribute to neurotransmission and synaptic plasticity. Polychlorinated biphenyl (PCB) exposure has been reported to alter intracellular Ca^2+^ levels and to interfere with normal neuronal dendritic growth and plasticity in a RyR-dependent manner (MIE L) [55].Thyroid hormone receptors α and β (THRα and THRβ) mediate the effects of thyroid hormones, while thyroperoxidase (TPO) and deionidase are involved in the biosynthesis/catabolism of thyroid hormones. Transtyretrin serum binding protein (TTR), monocarboxylate transporters 8 and 10, and the solute carrier organic anion transporter family member 1C14 (OATP1C1) are involved in the transportation of thyroid hormones at various levels [56]. Interference at any of these levels (MIEs G and Q-T) may lead to decreased thyroxine (T4) and thyroid hormones in the brain, and ultimately alter neurodevelopmental processes such as neuronal proliferation, apoptosis, migration, neurite outgrowth, and neuronal network connectivity [57,58], culminating in irreversible mental retardation and motor deficits [59]. It has been reported that PCBs induce activation of xenobiotic nuclear receptors, e.g., the constitutive androstane receptor (CAR) and the pregnane X receptor (PXR), which represent MIE P, leading to thyroid hormone disruption during cochlear development and potentially resulting in permanent auditory loss [60].The complexes of the respiratory chain play a pivotal role in neuronal and glial cell survival and cell death, as they regulate both energy metabolism and apoptotic/necrotic pathways. The interaction of xenobiotics with these enzymes can interfere in various ways with their normal functionality, e.g., inhibiting the production of ATP (MIE M) or interfering with the redox cycle (MIE N and O), with consequent increased production of ROS and oxidative stress. Oxidative stress contributes to a loss of function of hippocampal neural progenitor cells and a decline in learning and memory performance [4]. Moreover, the inhibition of NADH-quinone oxidoreductase (NADHOX) (MIE C) by pesticides or toxins (e.g., neurotoxin 1-methyl-4-phenyl-1,2,3,6- tetrahydropyridine, MPTP) has been reported to cause mitochondrial dysfunction and degeneration of dopaminergic neurons of the nigro-striatal area, with consequent motor deficits typical of Parkinson’s disease [61].Voltage-gated sodium channels (VGSC) are the primary molecules responsible for the control of the electrophysiological potentials of electrically excitable cells. Various isoforms exist, with isoforms 1, 2, 3, and 6 reported to be mainly expressed in the central nervous system [62]. Neurotoxic effects in mammals have been associated with the ability of some neurotoxicants (e.g., p,p’-DDT and pyrethroids) to bind to and disrupt VGSC (MIE U), with consequent behavioural effects [4,63].Ionotropic GABA receptors (GABAR) are ligand-gated ion channels which play important roles in inhibitory neurotransmission [64]. Interference with GABA signalling (MIE V) during development and after brain maturation is likely to cause such varied adverse outcomes as autism, mental retardation, epilepsy, and schizophrenia [59]. Chemically-induced epileptic seizures can be caused by the binding of neurotoxicants (e.g., barbiturates, benzodiazepines, and picrotoxin) to the active sites of the GABA receptor [65].

Data relative to each of the molecular targets identified in Table 3 were extracted from the ChEMBL database [66] using a protocol adapted from Bosc et al. [30]. ChEMBL Target IDs for each molecular target linked to an MIE are listed in Table 4. When available, only data relative to *Homo sapiens* were considered. For each target, only bioactivities with pChEMBL values were chosen. This term refers to all the comparable measures of half-maximal responses (molar IC50, XC50, EC50, AC50, Ki, Kd, potency and ED50) on a negative logarithmic scale [67]. Different pChEMBL thresholds to classify bioactivity values were evaluated. Ultimately, selected pChEMBL data were flagged as active or inactive based on a pChEMBL threshold of 5.0 (10 µM), providing datasets with a reasonable number of active samples for modelling. pChEMBL-like activities with standard relation “>“ or “≥“ (i.e., not associated with a precise activity value) were included as inactive. Only activities that were not flagged as potential duplicates, with no data_validity_comment and with an activity_comment that was not ‘inconclusive’, ‘undetermined’, or ‘not determined’ were considered. Endpoints characterised by few or no active compounds (i.e., thyroid hormone transporters, monocarboxylate transporters 8 and 10 and OATP1C1, NADH-cytochrome b5 reductase, deiodinase, and thyroperoxidase) were excluded from the modelling. For the modelling of NADH oxide reductase activity *Bos taurus* data were used, as human data were not available. pCheMBL data distributions were heavily skewed in the majority of cases towards positive values.

In order to prevent skew towards positive values, which is different from the natural distribution of biological data (i.e., few active, many inactive compounds), each dataset was further enriched with the chemicals included in the datasets of the remaining endpoints. These chemicals were treated as ‘decoys’ and assumed to be inactive. Due to the very large number of data available, AChE data were not used to enrich inactive samples of other endpoints to avoid the creation of datasets excessively unbalanced towards inactives. A semi-automated curation procedure [70] was applied to SMILES strings retrieved from ChEMBL in order to neutralise ionised chemical structures, remove counterions, and discard inorganics, organometallics, and mixtures. Removal of duplicate structures was carried out automatically at the InChI level. The entry with the maximum pChEMBL activity was selected in the case of duplicate structures in order to maximise the number of active samples.

Table 4 reports the final distribution of active and inactive chemicals for each dataset. The training sets for each of the modelled MIEs are available in the Appendix A.

### 4.2. QSARs for Molecular Initiating Events

ChEMBL datasets from Table 4 were used to develop 15 QSARs for molecular targets involved in the MIEs. Extended fingerprints (Daylight Chemical Information Systems, Inc., 2019) were calculated for each compound with a KNIME implementation [71] of the CDK toolkit (https://cdk.github.io/ (accessed on 7 March 2022)) and used as input for QSAR modelling. The BRF [72] implemented in KNIME was used for QSAR development. This technique artificially alters the class distribution in each tree. A sampling without repetition was made to select compounds, allowing all of the active compounds to always be selected together with an equal number of randomly selected inactive compounds from the training set in order to assure balancing between categories [73]. The number of trees in each BRF was set to 100.

Models were validated by splitting each data set into a training (80%) and a test set (20%) by applying a stratification sampling to the activity classes. The splitting procedure was repeated 100 times using different random splits, ensuring that each chemical in the datasets was included in the test set the same number of times in order to avoid bias due to the molecules present in the different sets. The performance using the test set was calculated for each iteration, then the final performance of each model was calculated by averaging the statistical parameters obtained using the test sets relative to each of the 100 iterations.

### 4.3. Thyroperoxidase (TPO) Modelling

As no data were found for TPO inhibition from ChEMBL, the QSAR model for predicting TPO inhibition proposed by Gadaleta et al. [68] was used. The model was developed from data related to the Amplex UltraRed-thyroperoxidase (AUR-TPO) assay. For positive hit-calls, only high selective inhibitors were used for the development of the model. These data were characterised by a demarcated separation of the AUR-TPO assay log IC20 value from confounding activities reported by a luciferase inhibition assay (flagging for non-specific enzyme inhibition) and a cytotoxicity assay. The QSAR was based on a BRF developed with the imbalance-learner and scikit-learn Python libraries [74] and based on 160 DRAGON descriptors [75] with a training set of 723 chemicals. Additional details on the predictive performance of the model can be found in [68].

### 4.4. Reactivity SMARTS

MIEs D (Binding and SH/SeH proteins involved in protection against oxidative stress) and H (Protein Adduct Formation) do not refer to an interaction with a specific receptor/enzyme; rather, they describe non-specific covalent binding to biological macromolecules (i.e., proteins.). Because this type of binding refers to the intrinsic reactivity of molecules, SMARTS compiled by Enoch et al. [69] describing electrophilic protein binding reactions (71 SMARTS) were used to account for the two MIEs. Chemicals matching at least one of the SMARTS were flagged as positive (1); otherwise, they were negative (0).

### 4.5. Neurotoxicity Data

Predictions using single QSARs for MIEs of neurotoxicity were evaluated for their capability to predict the neurotoxic potential of chemicals. Neurotoxicity data were retrieved from Kosnik et al. [76], who listed data for a total of 73 compounds (41 neuroactive and 32 non-neuroactive). This is a sub-selection of a list of the EPA’s ToxCast chemicals, previously tested by Strickland et al. [77] for their neural network function in vitro as measured on primary cortical cultures grown on microelectrode arrays and then subsequently retested to confirm the measured activities. SMILES were retrieved from the chemical name and CAS number using the semi-automated data retrieval and curation procedure from Gadaleta et al. [70]. Four compounds (three neuroactive and one non-neuroactive) were removed because they were mixtures, inorganics, and/or organometallics, leading to a final dataset of 69 chemicals.

The final list of 69 compounds along with their neurotoxic classification is reported in the Appendix A.

### 4.6. Neurotoxicity Modelling

The MIEs for the 69 chemicals with data for neuroactivity were predicted with the 15 BRFs developed from the entire datasets in Table 4 using the BRF model to predict AUR-TPO from Gadaleta et al. [68], and were profiled with the SMARTS for electrophilic activity compiled by Enoch et al. [69].

The predictions for the 69 chemicals from Kosnik et al. [76] were reported in the form of probabilities associated with predictions, and are shown in Appendix A. Probabilities ranged from 0 to 1, and in the case of BRF are the percentage of trees within the BRF returning a ‘positive’ prediction. As a consequence, probabilities higher than 0.50 flag for positive predictions, while probabilities lower than 0.50 flag for negative predictions. Predictions equal to 0.50 were considered “not classified”.

The predictions generated by the MIE models for the 69 chemicals were used as independent variables to develop new integrated QSAR models for predicting the neurotoxicity of chemicals.

Three different classifiers able to naturally handle a high number of independent variables were trained based on the neurotoxicity data. Five different settings were applied for the various classifiers, as implemented in KNIME [71].
K-Nearest Neighbours (k-NN) [78]: Euclidean distance was used to calculate the similarity between the target and the neighbours. K was varied from 1 and 9, with a step of 2.Random Forest (RF) [79]: the number of trees was varied from 50 and 250, with a step of 50.Multi-linear Perceptron–Artificial Neural Networks (NNET) [80,81]: one hidden layer was used, with the number of hidden neurons varied from 2 to 12 with a step of 2.

In order to verify the capability of MIE prediction to discriminate neuroactive and non-neuroactive compounds, QSARs based on MIEs were compared with other models developed with the same algorithms (kNN, RF and NNET) and different independent variables (i.e., extended structural fingerprints [82] and chemical descriptors). Chemical descriptors were calculated by means of DRAGON software [75,83]. The initial pool of descriptors calculated by DRAGON was pruned by constant and semi-constant values (standard deviation < 0.001). Descriptors having at least one missing value were also discarded. In the case of highly correlated descriptors (absolute pair correlation > 0.90), only the one with the highest number of correlated descriptors was retained, while the others were discarded. This procedure led to a final pool of 747 descriptors.

Model performance was evaluated by five-fold cross validation. Fold-splitting was performed by applying a stratified sampling of the neurotoxicity categories. For each classifier and each selection of parameters, the seed applied when performing the split was maintained; thus, the various folds were always the same. The splitting procedure was repeated 100 times using the same list of 100 random seeds for each combination of classifiers and parameters; then, statistics were collected for each iteration. Considering the variation of splits and parameters, a total of 500 iterations were performed for each of the three classifiers.

The same procedure was repeated in turns using MIE predictions, fingerprints, and DRAGON descriptors as independent variables, for a total of 1500 models developed. In the case of NNETs, DRAGON descriptors were preliminarily normalised in a range of 0–1, as NNETs are sensitive to the normalisation of independent variables.

Figure 2 summarises the entire procedure described above, including data extraction and curation, MIE modelling, and neurotoxicity modeling.

### 4.7. Evaluation of MIE Relative Importance

The relative importance of the various MIEs on the neurotoxicity predictions was evaluated in two ways.
(1)MIEs were iteratively removed, then QSARs for neurotoxicity were developed with the remaining features, as described in Section 4.6. BAs were averaged among the various iterations and compared to the reference values of models developed using all of the variables. A reduction in performance after the removal of a specific MIE flags a strong relationship between the excluded MIE and neurotoxicity. On the contrary, MIEs are considered less relevant if their exclusion does not vary or improve baseline performance.(2)Variable importance was calculated for each MIE within RF models. A score was calculated based on the attribute usage statistics in the RF for each descriptor by counting how many times it was selected for a split (#split) and at which rank (“level”; the first two levels were considered) among all available attributes (#candidates) in the trees of the ensemble:
Variable importance = #splits(level 0)/#candidates(level 0) + #splits(level 1)/#candidates(level 1)(1)

Variable importance calculated in this way was averaged among the various modelling iterations.

## 5. Conclusions

In the present manuscript, a new take on the traditional QSAR methodology was proposed to predict neurotoxicity by employing the biological information associated with chemicals (in the form of ligand-based predictions of MIE activation data) in place of the traditional structural data. The main advantage of this approach is that it can both return a prediction of the adverse outcome and provide insights into the specific mechanisms and molecular events that trigger toxicity. Emphasising the specific mechanisms of action behind neurotoxicity will increase the confidence of scientists and regulators in the predictions returned by these models. Having information about the activated molecular targets that are responsible for an apical effect may, in some cases, provide indications to chemists of possible modifications to the structure of hazardous chemicals, allowing for the designing of safe alternatives.

Despite their increasing usage, the application of the AOP framework in computational toxicology remains hampered by numerous and serious challenges. In general, an AOP is always a simplification of more complex and articulated biological pathways. Indeed, for certain biological processes, there are gaps in definitive knowledge of all responsible molecular determinants and mechanisms. In the case of neurotoxicity, there is a lack of understanding of all the MIEs involved in the alteration of downstream KEs as well as the occurrence of the AOs [4]. Several of the MIEs initially identified from AOPs were not included in the modelling presented here due to the shortage of data. Gaps in knowledge regarding chemical concentrations and time of exposure to trigger MIE/KEs prevent the development of quantitative approaches and limit the development of AOPs for adult and developmental neurotoxicity mainly to qualitative ones [4,27]. Considering the fact that the AOPs studied in this work are likely to be incomplete, the results described here are even more encouraging. Indeed, the future availability of more high-quality data is likely to improve the predictive capability of single QSARs for MIEs, while the future availability of more detailed AOPs and the inclusion of additional MIEs will complete the overall infrastructure, possibly leading to a more accurate and reliable prediction of apical endpoints. The incorporation of exposure and toxicokinetics, i.e., absorption (e.g., blood–brain barrier penetration), distribution, metabolism, and excretion data represent a possible additional improvement of the results presented herein [84].

## Figures and Tables

**Figure 1 ijms-23-03053-f001:**
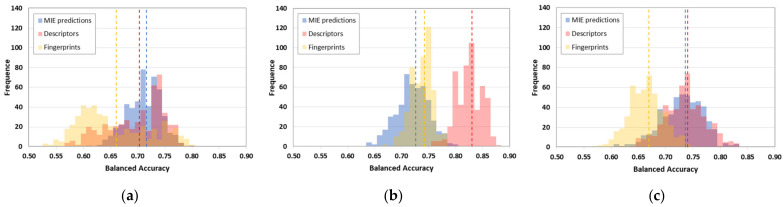
Distribution of balanced accuracies calculated among the various QSARs developed to predict neurotoxic potential. Balanced accuracies are grouped based on the algorithm used: (**a**) k-Nearest Neighbours; (**b**) Random Forest; (**c**) Neural Network. Blue bars refer to models developed based on MIE predictions, red bars refer to models based on DRAGON descriptors, and yellow bars refer to models based on Extended Fingerprints. Dashed lines indicate the mean accuracy value achieved by each group of models.

**Figure 2 ijms-23-03053-f002:**
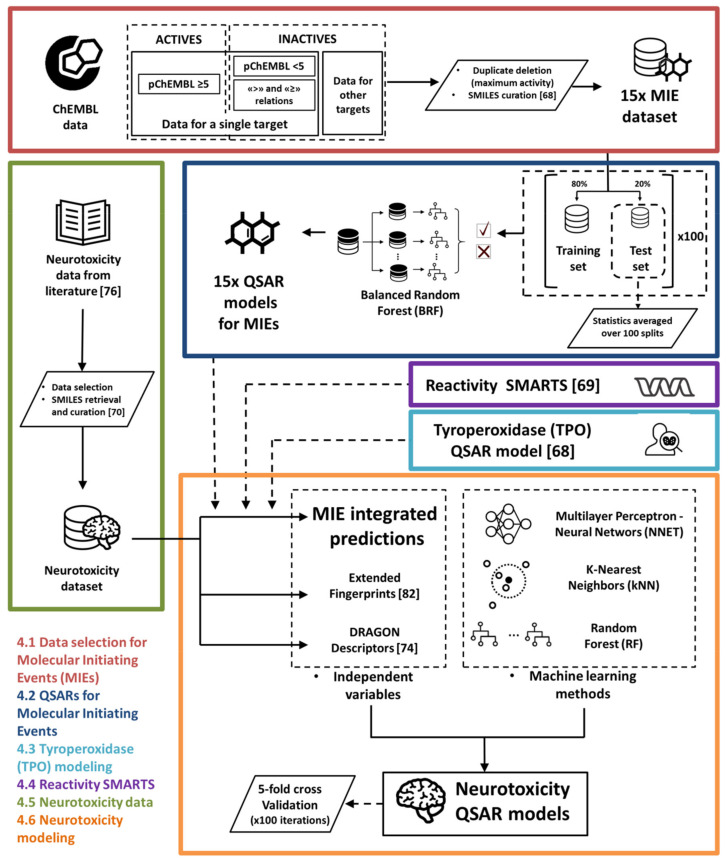
Modeling workflow. The colours of the various blocks refer to the paragraph in Materials and Methods that describes the specific steps of the workflow. Data from ChEMBL for 15 targets relevant for the MIEs of neurotoxicity (red) were classified based on the threshold pChEMBL = 5; negative samples were enriched with data using “>” and “≥” qualifiers and with chemicals from other MIE data that were treated as decoys. QSARs for MIEs (blue) were developed from these datasets using the BRF method. Datasets were iteratively partitioned into training and test sets and their external performance was calculated as the average of the various iterations; then, the models were retrained on the whole datasets. Neurotoxicity data (green) were retrieved from [76] and curated at the SMILES level. Predictions from the thyroperoxidase model (violet) by [72] and reactivity SMARTS (cyan) by [75] were combined with the predictions from the 15 MIE modes and used as independent variables to develop neurotoxicity QSAR models (orange). kNN, RF, and NNET were used to develop models. The use of MIE predictions as independent variables was benchmarked with fingerprints and DRAGON descriptors; then, the performance of the obtained models was compared with five-fold cross validation.

**Table 1 ijms-23-03053-t001:** External validation of QSAR models for MIEs based on ChEMBL data. For each MIE predicting QSAR the average number of true positives (TP), false positives (FP), true negatives (TN), and false negatives (FN) were reported. The metrics for evaluating the predictivity of the models were sensitivity (SEN), specificity (SPE), balanced accuracy (BA), Matthew’s correlation coefficient (MCC) and area under the ROC curve (AUC). Performance is the average of metrics obtained over 100 different training-test splits.

MIE	TP	FP	TN	FN	SEN	SPE	BA	MCC	AUC
AChE	555.6	63.0	887.2	68.0	0.89	0.93	0.91	0.83	0.96
AMPAR	13.4	20.0	651.0	1.2	0.92	0.97	0.94	0.60	0.99
CAR	9.0	6.8	668.6	1.2	0.88	0.99	0.94	0.72	0.95
CYP2E1	4.0	35.2	645.2	1.0	0.80	0.95	0.87	0.29	0.90
GABAR	20.0	11.0	649.0	6.0	0.77	0.98	0.88	0.69	0.96
KAR	4.4	17.4	663.0	0.6	0.88	0.97	0.93	0.42	0.97
NADHOX	15.2	4.4	665.4	0.4	0.97	0.99	0.98	0.87	1.00
NIS	11.0	0.6	673.4	0.4	0.97	1.00	0.98	0.96	1.00
NMDAR	50.0	27.8	604.4	3.4	0.94	0.96	0.95	0.75	0.98
PXR	35.8	35.8	601.8	13.0	0.73	0.94	0.84	0.57	0.92
RYR	11.0	0.6	673.6	0.2	0.98	1.00	0.99	0.96	0.99
THRα	60.0	23.4	599.2	2.8	0.96	0.96	0.96	0.81	0.99
THRβ	110.2	37.8	500.0	38.4	0.74	0.93	0.84	0.67	0.93
TTR	14.8	44.0	624.0	3.8	0.80	0.93	0.87	0.42	0.94
VGSC	28.4	12.8	639.2	5.0	0.85	0.98	0.92	0.76	0.97

**Table 2 ijms-23-03053-t002:** Performance of the three classifiers (kNN, RF, NNET) using MIE predictions, chemical descriptors, and extended fingerprints as independent variables. For each method, the average number of true positives (TP), false positives (FP), true negatives (TN) false negatives (FN) and not classified (NC) were reported. The metrics to evaluate the predictivity of the models were sensitivity (SEN), specificity (SPE), balanced accuracy (BA), Matthew’s correlation coefficient (MCC), and area under the ROC curve (AUC). Performance is the average of five-fold cross-validation results obtained over 500 iterations (100 fold-splitting procedures and five parameter combinations).

Classifier	Variable	TP	FP	TN	FN	NC	SEN	SPE	BA	MCC	AUC
K-NN	MIE predictions	30.5	11.4	19.6	7.5	0.0	0.80	0.63	0.72	0.44	0.76
	Descriptors	29.5	11.5	19.5	8.5	0.0	0.78	0.63	0.70	0.41	0.76
	Fingerprints	14.6	2.3	28.5	22.2	1.4	0.40	0.92	0.66	0.37	0.75
MLP-NNET	MIE predictions	29.4	9.4	21.6	8.6	0.0	0.77	0.70	0.74	0.47	0.78
	Descriptors	30.2	9.8	21.2	7.8	0.0	0.79	0.68	0.74	0.48	0.79
	Fingerprints	28.1	12.4	18.6	9.9	0.0	0.74	0.60	0.67	0.34	0.69
RF	MIE predictions	31.1	11.6	19.2	6.4	0.7	0.83	0.62	0.73	0.47	0.77
	Descriptors	32.9	6.4	24.4	4.9	0.4	0.87	0.79	0.83	0.66	0.91
	Fingerprints	32.9	11.9	18.8	4.8	0.5	0.87	0.61	0.74	0.51	0.80

**Table 3 ijms-23-03053-t003:** Molecular Initiating Events associated with Developmental Neurotoxicity, adapted from Spînu et al. [45] and Li et al. [27].

ID	MIE	Target	Reference
A	Binding of agonist, Ionotropic glutamate receptors	Glutamate [NMDA] receptor	[45]
A	Binding of agonist, Ionotropic glutamate receptors	Glutamate receptor ionotropic kainate	[45]
A	Binding of agonist, Ionotropic glutamate receptors	Glutamate receptor ionotropic AMPA	[45]
B	Binding of antagonist, NMDA receptors	Glutamate [NMDA] receptor	[45]
C	Binding of inhibitor, NADH-ubiquinone oxidoreductase (complex I)	Mitochondrial complex I (NADH dehydrogenase)	[45]
D	Binding, SH/SeH proteins involved in protection against oxidative stress	Aspecific1	[45]
E	CYP2E1 Activation	Cytochrome P450 2E1	[45]
F	Inhibition, Na^+^/I^−^ symporter (NIS)	Sodium/iodide cotransporter	[45]
G	Thyroperoxidase, Inhibition	Thyroid peroxidase ^1^	[45]
H	Protein Adduct Formation	Aspecific ^2^	[45]
I	Binding of inhibitors to acetylcholinesterase (AChE)	Acetylcholinesterase	[27]
L	Binding of non-dioxin-like polychlorinated biphenyls with ryanodine receptor (RyR)	Ryanodine receptors 1, 2 and 3	[27]
M	Interaction uncouplers with oxidative phosphorylation	Aspecific ^3^	[27]
N	Binding of redox cycling chemicals with NADH-quinone oxidoreductase	Mitochondrial complex I (NADH dehydrogenase)	[27]
O	Binding of redox cycling chemicals with NADH cytochrome b5 reductase	NADH-cytochrome b5 reductase	[27]
P	Xenobiotic nuclear receptor activation	Pregnane X receptor	[27]
P	Xenobiotic nuclear receptor activation	Nuclear receptor subfamily 1 group I member 3 (Constitutive Androstane Receptor)	[27]
Q	Interference with thyroid serum binding protein	Transthyretin	[27]
R	Deiodinase inhibition	Deiodinase ^4^	[27]
S	Thyroid receptor binding	Thyroid hormone receptor beta	[27]
S	Thyroid receptor binding	Thyroid hormone receptor alpha	[27]
T	Thyroid hormone transporter interference	Monocarboxylate transporter 8 ^4^	[27]
T	Thyroid hormone transporter interference	Monocarboxylate transporter 10 ^4^	[27]
T	Thyroid hormone transporter interference	Solute carrier organic anion transporter family member 1C1 ^4^	[27]
U	Binding of pyrethroids to voltage-gated sodium channels (VGSC)	Sodium channel protein type N alpha subunit	[27]
V	Binding of antagonist to γ-aminobutyric acid receptor GABAAR	GABA-A receptor; alpha-1/beta-2/gamma-2	[27]

^1^ No data found in ChEMBL, QSAR from Gadaleta et al. [68] was used. ^2^ Replaced with the use of reactivity SMARTS [69]. ^3^ No specific targets, not considered for modelling. ^4^ No data found in ChEMBL, not considered for modelling.

**Table 4 ijms-23-03053-t004:** List of endpoints modelled using ChEMBL data. For each endpoint, the reference MIE, ChEMBL ID relative to the molecular target, species, and composition of the Training and Test sets are reported; ACT is the number of active compounds, while INA is the number of inactive compounds.

Target	Code	CheMBL ID	Species	MIE	ACT	INA
Acetylcholinesterase	AChE	CHEMBL220	Human	I	3076	4793
Glutamate receptor ionotropic AMPA	AMPAR	CHEMBL2096670	Human	A	73	3355
Nuclear receptor subfamily 1 group I member 3 (Constitutive Androstane Receptor)	CAR	CHEMBL5503	Human	P	51	3377
Cytochrome P450 2E1	CYP2E1	CHEMBL5281	Human	E	25	3402
GABA-A receptor; alpha-1/beta-2/gamma-2	GABAR	CHEMBL2095172	Human	V	129	3298
Glutamate receptor ionotropic kainate	KAR	CHEMBL2109241	Human	A	25	3402
Mitochondrial complex I (NADH dehydrogenase)	NADHOX	CHEMBL614865	Bos taurus	C, N	78	3349
Sodium/iodide cotransporter	NIS	CHEMBL2331047	Human	F	56	3371
Glutamate [NMDA] receptor	NMDAR	CHEMBL2094124	Human	A, B	267	3161
Pregnane X receptor	PXR	CHEMBL3401	Human	P	244	3188
Ryanodine receptors ^1^	RYR	CHEMBL2062 CHEMBL4403 CHEMBL1846	Human	L	56	3371
Thyroid hormone receptor alpha	THRα	CHEMBL1860	Human	S	311	3116
Thyroid hormone receptor beta	THRβ	CHEMBL1947	Human	S	728	2704
Transthyretin	TTR	CHEMBL3194	Human	Q	93	3340
Sodium channel protein type N alpha subunit ^2^	VGSC	CHEMBL1845 CHEMBL4187 CHEMBL5163 CHEMBL5202	Human	U	167	3260

^1^ All the three isoforms of RYR were considered. ^2^ Isoforms 1, 2, 3 and 6 were considered.

## Data Availability

The data presented in this study are available in the Appendix A.

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
