# Peer review of "Prediction of the Neurotoxic Potential of Chemicals Based on Modelling of Molecular Initiating Events Upstream of the Adverse Outcome Pathways of (Developmental) Neurotoxicity"

_ijms, 2022, doi:10.3390/ijms23063053_

Round 1

Reviewer 1 Report

This is an interesting paper definitely worth publishing, but it needs correction of a number of errors that probably occurred in its writing and a small number of clarifications. We are also concerned by the unrealistic performance of the balanced random forest algorithm used, because it uses simple cross-validation on artificial data sets, and as discussed below, in our experience the performance on the artificial data sets tend to be indeed unrealistically high, but on external data sets is as disappointing as building models on unbalanced data.

Lines 247-248: “MIEs linked to neurotoxicity were identified from the AOP networks [44] published 

by Spinu et al. [45] and Li et al. [27].” Actually reference 45 refers to a publication by Carlson, L. M. et al., whereas reference 27 refers to a publication by Gadaleta D et al. The authors should check again and correct the references mentioned here. This is also valid for Table 3. I am afraid that the other references throughout the paper could also be misnumbered.

The PCB (polychlorinated biphenyls) abbreviation is first introduced on line 290, but it is explained on line 303 (the explanation should accompany its first use).

The four Table 3 footnotes are not identifiable within the Table (it is not clear to which row they refer to).

Line 348: the pChEMBL threshold of 5 used to classify compounds in “actives “ and “inactives” should be justified, as it seems arbitrarily chosen. Our understanding is that a threshold of 5 corresponds to a 10 micromolar level, which could be a reasonable one, but it is no less arbitrary.

Lines 348-349: “pChEMBL-like activities with standard relation ">" or ">=" (i.e. not associated to a precise activity value) were included as inactive.” This is often true, but not always. Sometimes the ">" or ">=" relation refers to a low concentration level (e.g. > 100 nM), and in such cases the relationship is too vague to be of any relevance, and that compound should rather be eliminated.

Lines 366-367: this selection approach should be justified against other options (such as the use of median or mean values).

388-394: Apparently the authors used a repeated (n=100) holdout cross-validation, which is probably biased. A double (nested) cross-validation would be highly preferrable.

Lines 454-460: Considering the very small number of compounds (n=69), we are not convinced that the approach used by the authors has avoided overfitting, and the results are likely to be biased.  

It is not clear to us if any feature selection method was used or all the features were included in the QSAR models built (including, in the case of Dragon descriptors or fingerprints, possibly several thousands of features). It is also not clear whether any hyperparameter tuning was used within the modeling process. No information is also provided on the software used to build the neurotoxicity QSAR models (unlike those used for the MIE models).

For both categories of models, the authors limited themselves to different forms of repeated cross-validation, but no true external testing set was used, and we are much afraid that his approach is prone to bias (this is confirmed by the very high performance of the models, including AUCs of 1.00). In our view, the use of the phrase „external validation” in this context is rather inappropriate and it seems likely that there is a certain degree of overfitting in the process. From our experience, the use of artificially enriched databases lead to extremely good performances (even in nested cross-validation) on those artificial datasets, but the results on true external data sets were very disappointing. We have observed this with over-sampling, undersampling and ROSE. Therefore, we tend to be very sceptical of the results with 99% (or similarly high) balanced accuracy, and we tend to believe that these are due to the BRAF algorithm using synthetic data. The results from Table 2 are much more realistic and in line with what one usually gets from real QSAR models. If our understanding is correct, for those model, no  balancing of the data sets was used in the modelling and this could explain the large discrepancy in performance. I would suggest the authors to leave outside a small number of compounds to test the models from Table 1 on real external data sets in order to have a realistic assessment of those models (e.g. they could leave aside 20% of the compounds and then test the BRF models on those external data sets and assess their performance).

Lines 175-176: it was difficult for us to understand exactly this sentence, because we could not find those BA values in Table 2.

Reviewer 2 Report

This manuscript describes a screening method that integrates multiple Quantitative Structure-Activity Relationship (QSAR) models and models the MIEs of existing AOP networks of developmental and adult/ageing neurotoxicity to predict neurotoxicity.

A few suggestions for the authors.

  1. The organization of the manuscript is not very reasonable. Table 1 lists the MIEs without any explanation and references. A few explanations about these MIEs present from line 146 to line 157. And then Table 3 contains the information for those MIEs followed by long explanations.
  2. Figure 1 does not have a reasonable description.
  3. Some of the abbreviations are shown with out the full description, which can be very difficult for the readers to follow.
  4. In the session 4. Materials and Methods, no clear description of the integrated QSAR, no clear workflow.

Round 2

Reviewer 1 Report

I am still skeptical on the bias of the assessments, but the paper has definitely improved and it could be made public.

Author Response

We thank the reviewer for considering our manuscript suitable for publication.

Reviewer 2 Report

The revised manuscript has improved a lot. Only suggestion is maybe the authors could provide a list of abbreviations at the end of manuscript.

Author Response

We thank the reviewer for his positive comments. A list of the abbreviations used has been now included at the end of the manuscript as suggested.